# Generation of Lasso Peptide-Based ClpP Binders

**DOI:** 10.3390/ijms23010465

**Published:** 2021-12-31

**Authors:** Imran T. Malik, Julian D. Hegemann, Heike Brötz-Oesterhelt

**Affiliations:** 1Department of Microbial Bioactive Compounds, Interfaculty Institute of Microbiology and Infection Medicine, University of Tübingen, 72076 Tübingen, Germany; imiq23@gmail.com; 2Helmholtz Centre for Infection Research (HZI), Helmholtz Institute for Pharmaceutical Research Saarland (HIPS), Saarland University Campus, 66123 Saarbrücken, Germany; 3Cluster of Excellence “Controlling Microbes to Fight Infection”, University of Tübingen, 72076 Tübingen, Germany

**Keywords:** Clp protease, Clp ATPase, lasso peptide, epitope grafting, mutagenesis, bioengineering, acyldepsipeptide, ADEP, natural product, proteolysis

## Abstract

The Clp protease system fulfills a plethora of important functions in bacteria. It consists of a tetradecameric ClpP barrel holding the proteolytic centers and two hexameric Clp-ATPase rings, which recognize, unfold, and then feed substrate proteins into the ClpP barrel for proteolytic degradation. Flexible loops carrying conserved tripeptide motifs protrude from the Clp-ATPases and bind into hydrophobic pockets (H-pockets) on ClpP. Here, we set out to engineer microcin J25 (MccJ25), a ribosomally synthesized and post-translationally modified peptide (RiPP) of the lasso peptide subfamily, by introducing the conserved tripeptide motifs into the lasso peptide loop region to mimic the Clp-ATPase loops. We studied the capacity of the resulting lasso peptide variants to bind to ClpP and affect its activity. From the nine variants generated, one in particular (12IGF) was able to activate ClpP from *Staphylococcus aureus* and *Bacillus subtilis*. While 12IGF conferred stability to ClpP tetradecamers and stimulated peptide degradation, it did not trigger unregulated protein degradation, in contrast to the H-pocket-binding acyldepsipeptide antibiotics (ADEPs). Interestingly, synergistic interactions between 12IGF and ADEP were observed.

## 1. Introduction

Lasso peptides belong to the natural product superfamily of ribosomally synthesized and post-translationally modified peptides (RiPPs) [1,2,3,4]. They are characterized by their lariat knot-like [1]rotaxane topology (Figure 1A) and are renowned for their high proteolytic and in many cases also thermal stability [4].

Like for all RiPPs, their biosynthesis starts with the ribosomal assembly of a genetically encoded precursor [1,2,3,4]. Processing of the precursor into a mature lasso peptide requires the action of three proteins (Figure 1B) [2,3,5,6,7,8,9,10,11,12]: (1) a RiPP recognition element (RRE) recognizes a conserved motif in the N-terminal leader region of the precursor peptide, binds to it and mediates contact to (2) a cysteine protease with homology to transglutaminases. The leader peptidase then releases the C-terminal core peptide by proteolysis, and thereby allows (3) a lasso cyclase with homology to asparagine synthetases to act on it. In the last step of the biosynthesis, the lasso cyclase activates the sidechain of an aspartate or a glutamate at positions 7, 8, or 9 in an ATP-dependent manner and finally catalyzes a condensation reaction between the activated carboxylic acid moiety and the N-terminal α-amine. The macrocyclization reaction is likely accompanied by a prefolding mechanism that allows macrolactamization to occur around the linear C-terminal tail, thereby entrapping it in the macrocycle. Stabilization of this fold is accomplished by placement of amino acids with bulky sidechains above and below where the tail threads the ring [2,3,4]. 

In the so far characterized lasso peptide biosynthetic gene clusters, RREs and peptidases can either be found as discrete proteins (B1 and B2, respectively) or as a single bifunctional polypeptide (B), while the lasso cyclase (C) is typically not linked to other proteins. Recently, a novel subtype of lasso peptide gene clusters was described, where the encoded leader peptidase is fused to the N-terminus of an ABC-transporter (B2/D) [12]. Gene clusters are known to encode either one or several precursor peptides (A). It was furthermore demonstrated that the RRE binding of the precursor is a prerequisite for proteolysis to occur [6,7,8,9,10].

Microcin J25 (MccJ25) is the most well-studied lasso peptide (Figure 2A). It was discovered in 1992 and exhibits antibiotic activity against Enterobacteriaceae [8,13]. Its primary mode of action is the inhibition of the Gram-negative RNA polymerase through blocking of the nucleoside triphosphate (NTP) uptake channel [14,15,16,17,18]. MccJ25 enters its target cells through hijacking the TonB-dependent siderophore receptor FhuA [8,19,20] which explains its narrow activity spectrum. MccJ25 originates from *Escherichia coli* AY25 [13] and thus can be produced very well heterologously in standard *E. coli* expression strains using the pTUC202 plasmid (Figure 2B) [21,22]. An ABC transporter (McjD) expressed on this plasmid mediates the export of MccJ25 in the extracellular space and hence provides self-immunity to the producing strain [21,23,24,25].

MccJ25 is exceptionally stable, even for lasso peptide standards. It was shown to withstand autoclaving at 120 °C [4,13]. Even hydrolysis of a peptide bond in the loop region (for example between Phe10 and Val11, as induced by thermolysin) does not cause ring and tail to dissociate as the steric interactions are sufficient to stabilize the resulting [2]rotaxane in both gas phase and in solution [26,27,28,29]. Due to its high yields, high stability, and its loop region of above average length, MccJ25 is very well suited for epitope grafting, i.e., the incorporation of small peptide epitopes that mediate specific biological activities. For example, by incorporation of the RGD integrin recognition epitope, variants of MccJ25 were generated that exhibited a high affinity for binding to the αvβ3 integrin receptor [22].

We hypothesized that the Clp protease complex might also be an interesting target for lasso peptide epitope grafting. The Clp protease is a highly conserved two-component system, which consists of the proteolytic ClpP as well as a regulatory Clp-ATPase [30]. In many bacterial species, the Clp protease was shown to be important for protein homeostasis and quality control, cell differentiation, virulence regulation, and stress management [31]. In pathogenic bacteria, the Clp protease system serves as a promising target for antivirulence and antibacterial treatment [31,32].

ClpP is shaped like a barrel and is composed of two heptameric rings that stack upon each other (Figure 3A). At the top and the bottom of the barrel are narrow entry pores for substrate translocation [33]. Clp-ATPases, like ClpA, ClpX, or ClpC, are responsible for the recognition of substrates and their translocation into the catalytic cavity of ClpP, where the degradation takes place [30]. In the proteolytically active complex, the two components form a larger stack with the hexameric ring-shaped Clp-ATPases sitting on one or both sides of the ClpP barrel (Figure 3A) [34,35,36]. Without the supporting action of a Clp-ATPase, ClpP alone does not degrade proteins because the access to the proteolytic core through the narrow entrance pores is limited to small peptides [37]. Upon a degradation signal, the Clp-ATPases bind the substrate, establish contact to ClpP, and actively unfold and translocate the substrate through the entrance pores. Of particular relevance in this interaction is a flexible loop that protrudes from each Clp-ATPase protomer and carries a conserved (L/I/V)-G-(F/L) hydrophobic tripeptide binding motif at its tip (Figure 3B) [38]. With the help of modelling and mutational studies, hydrophobic binding pockets (H-pockets) at the apical surface of ClpP were identified that serve as anchors for the flexible loop motifs of Clp-ATPases (Figure 3A) [39,40].

We were interested in investigating whether grafting of the Clp-ATPase tripeptide binding motifs onto the loop region of MccJ25 would yield novel MccJ25 variants able to bind to and affect the activity of ClpP tetradecamers. An inspiration for this approach came from the acyldepsipeptide antibiotics (ADEPs) [41]. ADEP congeners activate and deregulate ClpP by binding to the H-pockets, which triggers a conformational shift in the ClpP barrel [42,43]. Binding increases the pore diameter and allosterically enhances catalytic activity, which in turn leads to an unregulated digest of essential cytosolic proteins and thereby cell death [44,45,46,47].

In addition, an ADEP competitively disrupts the functional interactions between ClpP and Clp-ATPases and hence abrogates all natural functions of the Clp protease [48,49]. Thereby, ADEPs exhibit potent antibacterial activities against a broad range of Gram-positive bacteria in vitro and in vivo [41,50]. The ADEP structure consists of a macrolactone core and an N-acylphenylalanine sidechain (Figure 3B) [51]. Analysis of several ADEP fragments revealed that the latter structure is indispensable for activity, thus forming the pharmacophore [52]. The macrolactone core, while being inactive on its own, increases potency of the sidechain by establishing additional contacts to the H-pocket. Intriguingly, the N-acylphenylalanine fragment necessary for activity closely resembles the tripeptide binding motifs within the protruding loops in Clp-ATPases (Figure 3B), supporting the notion of the ADEP sidechain as an IGF loop mimic [42].

With the aim to generate lasso peptides that reflect the Clp-ATPase loop conformation, we introduced various tripeptide binding motifs into the loop region of MccJ25. We reasoned that the loop of a lasso peptide might be converted into a suitable mimic of the Clp-ATPase binding loop and that these peptides present an ideal basis for offering the conserved binding epitope to ClpP without the higher structural organization of a Clp-ATPase hexamer. We furthermore assumed that the high proteolytic stability of lasso peptides would protect them against degradation by ClpP.

The loop of MccJ25 contains a stretch of five residues, Val11-Gly12-Ile13-Gly14-Thr15, that would not only allow the introduction of common Clp-ATPase binding epitopes (IGL, IGF, and VGF) without too drastic changes in the peptide sequence but would also position them well for the presentation to ClpP.

Here, we present the production and isolation of a total of nine MccJ25 variants, which carry either of the aforementioned three epitopes at different positions of their loop regions (Figure 3C). We demonstrate that some of our lasso peptides are indeed able to interact with ClpP and can slightly affect ClpP activity, which was not observed for wild type (WT) MccJ25. Unexpectedly, our results also show that the active MccJ25 variants do not seem to primarily interact with the H-pockets. Instead, these variants apparently bind to another, yet unidentified, region of the ClpP tetradecamer, which allows synergistic behavior when using lasso peptide-based ClpP binders and ADEPs in parallel. Whereas the overall activity of our novel lasso peptide-based ClpP binders is still low, this conceptual study nonetheless showcases the potential lasso peptides hold for targeting the Clp protease system. 

## 2. Results

### 2.1. Construction of MccJ25 Variants with Canonical Clp-ATPase Tripeptide Binding Motifs

MccJ25 variants were generated with the intent to imitate the tripeptide binding motif of cognate Clp-ATPases. Therefore, the three common tripeptide motifs (IGL, IGF and VGF) found in the respective Clp-ATPases from the model organisms *E. coli* (ClpA), *S. aureus*, *B. subtilis*, and *Helicobacter pylori* (ClpX), were introduced each at three positions within the loop of MccJ25 (Figure 2A and Figure 3C). Thus, a set of nine lasso peptide variants was generated, incorporating IGL, IGF, or VGF starting at either position 11 (11IGL, 11IGF, 11VGF), position 12 (12IGL, 12IGF, 12VGF), or position 13 (13IGL, 13IGF, 13VGF) (Figure 3C).

Production yields strongly depended on the position of the incorporation and relied on how drastically the primary structure of MccJ25 was altered (Figure 3C). The best yields (~18–25 mg/L culture) were obtained for exchanges starting at position 11 (change of VGI to IGL, IGF, or VGF) and variants with exchanges from position 13 and onwards (change of IGT to IGL, IGF, or VGF) were still produced in good amounts (~5–7 mg/L). However, only low yields (~0.5 mg/L) were obtained for variants, where the tripeptides were introduced at positions 12–14 (change of GIG to IGL, IGF, or VGF). In these cases, the small glycine residues were replaced with large hydrophobic amino acids, whereas the hydrophobic Ile13 was in return replaced with a glycine. 

While a previous systematic structure-activity-relationship study [53] on MccJ25 suggests that each of the individual exchanges at positions 11–15 performed in our study should be tolerated well, these findings demonstrate that the incorporation of multiple exchanges at once can show a cumulative detrimental effect with regards to enzymatic processing and in turn overall yields.

### 2.2. MccJ25 Variants Stabilize the Oligomeric State of the S. aureus ClpP Barrel

In order to assess the different MccJ25 variants, we first chose *S. aureus* ClpP (SaClpP) as a model peptidase and performed in vitro degradation assays with the fluorogenic model peptide substrate Suc-LY-AMC. Apo-SaClpP is purified as a stable tetradecamer and thus exhibits intrinsic peptidase activity that can be enhanced weakly by ADEP2 under established assay conditions in a HEPES buffer at pH 7 (Figure 4A) [43]. To investigate whether the MccJ25 variants can exert a stabilizing effect on the oligomeric structure of tetradecameric SaClpP, we conducted peptide degradation assays under destabilizing conditions employing a Tris buffer at pH 8. Under these conditions, peptidase activity of apo-SaClpP was almost completely abolished, but we noted that a substantial extent of the original activity (observed in the favorable HEPES buffer at pH 7) was retained in the presence of ADEP2 (Figure 4A). Gel filtration chromatography demonstrated that the detrimental effect of Tris buffer at pH 8 manifests in a breakdown of the oligomeric state of SaClpP (Figure 4B).

The full complex consisting of 14 ClpP protomers is roughly 316 kDa in molecular weight but disassembled in Tris buffer into three fractions: a predominantly heptameric fraction and lower oligomeric states in the dimer/trimer range. In the presence of ADEP, SaClpP remained in the active tetradecameric conformation even under unfavorable buffer conditions (Figure 4B). This experimental setup was then used to differentiate the MccJ25 variants regarding their effect on ClpP stability. In a panel that comprised WT MccJ25 and all of its nine variants, only the three variants carrying the canonical tripeptide motif at positions 12–14 were able to retain SaClpP activity in Tris buffer, most notably the 12IGF variant (Figure 4C). Gel filtration furthermore confirmed that 12IGF is able to maintain tetradecamer integrity in Tris buffer (Figure 4D). The stabilizing effect of 12IGF is concentration-dependent with an apparent affinity constant of 26 μM (Figure 4E). The apparent affinity constant was deduced from enzyme activity assays and represents the effector molecule concentration at half maximal stimulation of enzyme activity (the lower the apparent affinity constant, the higher the affinity). To exclude that 12IGF serves as a SaClpP substrate, we incubated a mixture of SaClpP and 12IGF under assay conditions for up to 260 min. LC-MS confirmed that 12IGF remained intact (Appendix A).

To test whether the most potent lasso peptide variant, 12IGF, was able to interfere with communication between SaClpP and the *S. aureus* model Clp-ATPase ClpX (SaClpX), we also performed an enhanced GFP (eGFP) degradation assay, where, in the presence of ATP, eGFP carrying the C-terminal ssrA degradation tag is unfolded and translocated into the SaClpP degradation chamber by SaClpX [54]. It was shown before that the SaClpX-dependent degradation is readily inhibited by small amounts of ADEP due to the strong competition of ADEP and ClpX for the H-pocket, which results in the abrogation of the interaction between the Clp-ATPase and ClpP (Figure 4F) [43]. ADEP (in contrast to an ATP-fueled Clp-ATPase) cannot activate ClpP to degrade eGFP. We used the same assay setup for 12IGF and observed that 12IGF interfered with the eGFP degradation rate of ClpXP only very slightly (Figure 4F). This was not unexpected, since we assumed that the affinity of six individual IGF motifs would not be able to compete with the overall higher avidity of six IGF motifs arranged in a favorable orientation in the SaClpX hexamers that are docking to ClpP simultaneously. Albeit weak, the inhibition of SaClpXP by 12IGF was nonetheless reproducible and concentration-dependent indicating an actual effect on the ClpX-ClpP interaction.

### 2.3. MccJ25 Variants and ADEPs Synergistically Stimulate B. subtilis ClpP Peptidase Activity

In previous studies, *B. subtilis* ClpP (BsClpP) had been purified to a large extent in the monomeric form [47], which was also the predominant conformation under our purification and assay conditions. In accordance with its predominantly monomeric state, it exhibited only marginal intrinsic peptidase activity and was strongly stimulated by the addition of ADEP. This stimulated peptidase activity is achieved first and foremost by the ADEP-induced assembly of the active tetradecameric ClpP complex.

When we tested the MccJ25 variants for their ability to stimulate BsClpP peptidase activity, stimulation could not be achieved with any of the variants. While in the case of SaClpP, 12IGF was able to stabilize an already assembled, active tetradecameric complex, it did not achieve the assembly of the BsClpP barrel from monomers. However, in the presence of ADEP2, and thereby fully formed and active BsClpP tetradecameric complexes, certain MccJ25 variants triggered a further increase in peptidase activity, with 12IGF again showing the highest potency (Figure 5A). The activation was concentration-dependent and required much higher overall 12IGF concentrations compared to ADEP2 with an apparent affinity constant of 60 μM (Figure 5C).

This finding led us to the hypothesis that while our lasso peptides are unable to induce the assembly of the active tetradecamer in BsClpP on their own, they can still exert an activating effect on a preassembled complex. To test this notion, we purified BsClpP under specific buffer conditions that favor tetradecamer assembly and subsequently collected the tetradecameric fraction from a gel filtration column. Activity assays were then performed with the tetradecameric fraction and the most potent lasso peptide, 12IGF. 

Indeed, 12IGF was able to reproducibly stimulate the peptidase activity of preformed BsClpP tetradecamers, thus confirming that 12IGF can activate a fully assembled BsClpP barrel (Figure 5B). The apparent affinity of 12IGF for BsClpP was unaffected by the absence or presence of ADEP2 (Figure 5C,D).

While peptidase activity is reliant only on an assembled tetradecameric complex and a functional catalytic triad, degradation of the loosely folded model protein substrate casein additionally requires the opening of the entrance pores as achieved by ADEP binding [47]. In contrast to ADEP, 12IGF was not able to activate either SaClpP or BsClpP for degradation of the fluorogenic protein substrate FITC-casein (Appendix A).

### 2.4. Lasso Peptide-Mediated Activation Is Not Inhibited by ADEPs

ADEPs bind ClpPs from different organisms with a comparatively high affinity in the low single digit micro-molar range [50]. For example, our SaClpP FITC-casein degradation assay showed an apparent affinity constant below 1 μM (Appendix A). 12IGF EC_50_ values were determined in SaClpP and BsClpP peptidase assays as 26 and 60 μM, respectively, and thus exceeded the EC_50_ values measured for ADEP by one to two orders of magnitude (Figure 4E and Figure 5C,D). 

Based on these findings, we expected ADEP to readily displace 12IGF when competing for the same binding site. Therefore, BsClpP competition assays were performed with an increasing concentration of ADEP2, while the 12IGF concentration was kept constant. As a readout, we monitored the surplus of BsClpP peptidase activity that we had observed in the presence of the 12IGF/ADEP2 combination in relation to ADEP2 alone (Figure 5A).

Unexpectedly, the additional stimulation of BsClpP by 12IGF in the presence of ADEP2 remained constant even at a very high ADEP2 excess (100-fold *K*_app_ for ADEP; Figure 6A). Interestingly, while 12IGF and ADEP2 co-stimulated the catalytic peptidase activity of BsClpP, where the tetradecamer first had to be assembled, the lasso peptide reduced the activating potential of ADEP2 in the case of the already pre-assembled SaClpP tetradecamer. SaClpP peptidase activity reproducibly decreased with rising ADEP2 concentrations in the presence of 12IGF (Figure 6B). The reason for this effect is not known. Furthermore, ADEP2 was not able to reduce the effect of the lasso peptide, even when titrated two orders of magnitude beyond its *K*_app_ for SaClpP. Both of these findings establish a lack of competitivity between ADEP2 and 12IGF and suggest a binding mechanism for 12IGF that is different from ADEP2. Clearly, a high excess of a potent H-pocket binder does not eradicate the 12IGF effect. 

This came as a surprise since the MccJ25 variants had been designed specifically to mimic the Clp-ATPase IGF-loop binding. Based on the notion that only the H-pocket would serve as a docking point for the IGF motif, we had expected that 12IGF would be limited to the same binding sites as ADEP2. This unexpected finding therefore has to be addressed in future studies that will focus on investigating the concrete binding interactions of 12IGF with ClpP in more detail. However, such in-depth binding interaction studies are out of the scope of this proof-of-concept study.

## 3. Discussion

In this study, we set out to generate lasso peptide-based ClpP binders based on the structural similarity of the loop region of MccJ25 with the protruding (L/I/V)-G-(F/L)-loops of Clp-ATPases. To accomplish this goal, the conserved hydrophobic tripeptide binding motifs from Clp-ATPases were incorporated into our lasso peptide scaffold. We successfully generated and isolated nine novel MccJ25 variants and tested them for activity in assays with the ClpP proteins from *S. aureus* and *B. subtilis*.

From these nine variants, the ones carrying tripeptide binding motives at positions 12–14 exhibited the strongest effects, with 12IGF showing the highest potency. By comparing the effects of 12IGF with those of ADEP2, parallels, but also striking differences, can be seen. Like ADEP2, 12IGF affects ClpPs from different species. 12IGF stabilizes the tetradecameric state of ClpP and can stimulate ClpP catalytic activity. However, 12IGF lacks the capacity of ADEPs to assemble the primarily monomeric BsClpP and requires preassembled tetradecameric BsClpP to exert catalytic activation. 

Apparent affinity displayed by 12IGF was one to two orders of magnitude lower than for ADEPs. The low affinity is not surprising as even the fully assembled ClpX hexamer had displayed weaker affinities than ADEP2 [43]. ClpX carries six coupled IGF motifs in a rather favorable orientation. Hence, we expected the individual tripeptide motifs presented by the MccJ25 variants to bind less strongly than ClpX because the latter benefits from the avidity of six binding sites.

Activation of SaClpP and BsClpP by lasso peptides was best achieved by incorporation of the IGF motif, which also occurs naturally in both SaClpX and BsClpX. Among the series of loop variants, the IGF motif performed best when starting from position 12 of the MccJ25 sequence, which places it right at the tip of the lasso peptide loop (Figure 2A and Figure 3C). Thus, we are confident that the effect we observe is in fact due to the respective tripeptide motif and how it is positioned rather than due to secondary effects, which is supported by the fact that neither WT MccJ25 nor several of the other mutated variants constructed in this study affected either of the ClpPs.

A previous structure-activity-relationship study on fragments of the ADEP structure showed that the *N*-acyldifluorophenylalanine moiety of ADEP is sufficient to exhibit residual bioactivity against *B. subtilis* ClpP, although its antibacterial activity dropped 500-fold compared to intact ADEP (MIC of 8 μg/mL compared to 0.016 μg/mL of the intact ADEP lead structure) [52]. The isolated macrolactone core did not show any activity on its own. *N*-acyldifluorophenylalanine also superimposes nicely with the resolved LGF tripeptide from a *H. pylori* crystal structure (PDB code 1UM8) substantiating the notion that the sidechain of ADEP is a Clp-ATPase IGF-loop mimetic [42,46,55]. Two recent cryo-co-crystal structures of the full ClpXP machine further confirmed that the IGF-loops and the ADEP sidechain occupy the same hydrophobic pockets [34,35,46,47,54].

Notably, even though the *N*-acylphenylalanine sidechain of ADEP occupies the same binding site as the IGF motif, its macrolactone backbone allows binding to a greater surface area of ClpP and thus establishes various additional hydrophobic contacts plus two hydrogen bonds which explains the documented higher binding affinity. Whether the fusion of the ADEP N-acyldifluorophenylalanine sidechain to its macrolactone core produces conformational changes that IGF loop-binding might not achieve, is still not fully resolved. The fact that ADEP binding triggers pore opening (from ca. 10–15 Å in apoClpP to ca. 20–30 Å at full ADEP occupancy) was demonstrated for a range of different bacteria species and human mitochondrial ClpP (for a recent review see [50]), while divergent observations have been made for ClpXP in two independent recent studies. In the cryo-EM structure of *E. coli* ClpXP in the process of substrate digestion, the entry pores were widened to 30 Å, while the cryo-EM structure of *Listeria monocytogenes* ClpXP1P2 displayed a narrow pore diameter comparable to apoClpP1P2 [34,35].

Compared to *N*-acyldifluorophenylalanine, a linear peptide carrying the IGF motif that was tested in a previous study was much inferior and showed very little activity [38]. This behavior might be due to an unfavorable conformation or a too high conformational flexibility of the linear peptide and it is not known if this behavior correlates with the binding properties of individual IGF loops in the natural context. It has been shown that cyclization of peptides can increase their binding affinity [22,56,57,58,59] as the rigidity introduced through cyclization reduces the conformational flexibility and thus limits the conformational space that has to be sampled before a topology fit for binding is realized.

In addition to this entropic effect, cyclization can also increase peptide stability in a physiological setting. These beneficial effects, increased rigidity and proteolytic stability, can also be accomplished through incorporation of a target sequence into a lasso peptide. As hoped, we saw that the IGF motif as part of a lasso peptide loop surpassed the poor apparent affinity of the linear peptide and we demonstrated that some of the engineered lasso peptides were capable of stabilizing ClpP’s tetradecameric structure as well as stimulating its catalytic function. 

Previously, it was shown that for catalysis, ClpP needs to adopt the tetradecameric extended conformation [43,60,61]. ADEP as well as ClpX induce this state in ClpP by exerting conformational control [42,43]. The observation that 12IGF also stimulates catalysis implies corresponding conformational control by the lasso peptide. However, as 12IGF did not facilitate casein degradation by ClpP, it seems to be unable to trigger sufficient pore opening for entry of the loosely folded protein into the ClpP lumen. However, this result is in clear contrast to ADEP, whose antibacterial activity depends on the pore opening and the resulting deregulated protein and polypeptide degradation in bacterial cells. 

We were surprised to see that 12IGF displayed no competition to ADEP2, neither for SaClpP nor BsClpP. In the case of BsClpP, 12IGF and ADEP2 displayed synergistic behavior with 12IGF providing additional activity independent of the ADEP2 concentration. In SaClpP, the combination of ADEP2 and 12IGF showed an inhibitory effect that could not be alleviated by applying ADEP2 in high excess. If 12IGF and ADEP2 exclusively shared the same docking site, we would expect ADEP2 excess to successfully displace all the 12IGF molecules from ClpP and exhibit an ADEP2-only activity. The fact that this behavior was not observed implies that besides our initial working hypothesis, 12IGF does not bind to the H-pocket (or at least not exclusively) but instead to a so far unidentified site of the ClpP complex. This unexpected, yet intriguing finding will provide the basis for future investigations of the interactions between 12IGF and ClpP on a molecular level. 

The data presented furthermore emphasize that the high proteolytic stability of lasso peptides can be utilized to generate novel kinds of ClpP binders that cannot be degraded by Clp proteases. While overall activity of this first generation of lasso peptide-based ClpP binders is still low, the rise of yeast- and phage-display techniques for RiPPs [58,59] might facilitate the generation and screening of large lasso peptide libraries to follow up on our initial findings and potentially yield more potent compounds.

## 4. Materials and Methods

### 4.1. Mutagenesis of mcjA

Mutagenesis of the *mcjA* gene in the MccJ25 production plasmid (pTUC202) was accomplished by using site-directed ligase-independent mutagenesis (SLIM) according to published protocols [22,62,63]. In short, a set of four primers was used for every mutant. The same two base primers (*mcjA*_SLIM_FP and *mcjA*_SLIM_RP; Table 1) that anneal to the regions flanking the *mcjA* sequence stretch coding for residues 9–18 of MccJ25 were used for every mutant. Another two primers unique for every mutant were employed, which carried the mutated codons for residues 9–18 as 5′ overhang on the base primer sequence (Table 1). For every mutation, two 50 μL PCRs were performed using Phusion DNA polymerase (New England Biolabs, Frankfurt am Main, Germany): One PCR using *mcjA*_SLIM_FP and the respective overhang reverse primer, the other PCR utilizing the respective overhang forward primer and *mcjA*_SLIM_RP. Of each reaction, 10 μL were analysed by agarose gel electrophoresis to check if the target DNA was amplified and, if successful, the remainder was treated with DpnI (New England Biolabs, Frankfurt am Main, Germany) to remove the template DNA (40 μL PCR sample + 4.6 μL CutSmart Buffer + 1.0 μL DpnI; 2 h at 37 °C; followed by DpnI-inactivation for 20 min at 80 °C). DpnI-treated samples were then hybridized. For this, 10 μL of each DNA sample were mixed with 10 μL of 5× hybridization buffer (750 mM NaCl, 125 mM Tris, 100 mM EDTA, pH 8.0) and 20 μL of ddH_2_O. The mixture was then incubated for 3 min at 99 °C, followed by three cycles of incubation for 5 min at 65 °C and 40 min at 30 °C. For transformation of the hybridized, circular DNA, 10 μL of each hybridization reaction were used and *E. coli* TOP10 cells carrying the target plasmids were selected for by plating on lysogeny broth (LB) agar plates containing 17 μg/mL chloramphenicol. Incorporation of the correct mutations was confirmed via dideoxy sequencing by GATC Biotech AG (Konstanz, Germany).

### 4.2. Production and Purification of MccJ25 Variants 

For production of the MccJ25 variants, M9 minimal medium (17.1 g/L Na_2_HPO_4_∙12 H_2_O, 3 g/L KH_2_PO_4_, 0.5 g/L NaCl, 1 g/L NH_4_Cl, 1 mL/L MgSO_4_ solution (2 M), 0.2 mL/L CaCl_2_ solution (0.5 M), pH 7.0; after autoclaving, 10 mL/L sterile glucose solution (40% *w*/*v*) and 2 mL/L 500× M9 vitamin mix (Table 2) were added) with 17 μg/mL chloramphenicol was inoculated 1:100 with 37 °C LB overnight cultures also containing 17 μg/mL chloramphenicol and then the M9 minimal medium cultures were shaken for 3 days at 37 °C in baffled flasks (600 mL medium per 2 L flask). Afterwards, the cells were harvested by centrifugation and the supernatant containing the lasso peptides was extracted by stirring with XAD16 resin (20 mL of a suspension of 50 g XAD16 resin with 200 mL ddH_2_O (=250 mL total volume) was added for every liter of culture volume) for 1 h at RT. Next, the resin was collected on filter paper in a funnel, washed three times with 5 mL of ddH_2_O and then eluted in a stepwise manner with a total volume of 100 mL of MeOH per liter of original culture volume. The supernatant extract was dried under reduced pressure at 40 °C and resuspended in 8 mL of 50% MeOH in ddH_2_O. The resuspended extracts were cleared by centrifugation and subsequent sterile filtration. For testing if the target lasso peptides were produced, 100 μL of each extract was applied to high-resolution LC-MS employing a 125/2 Nucleosil 300-8 C18 column (Macherey-Nagel, Munich, Germany) that was connected to a microbore 1100 HPLC system (Agilent) and an LTQ-FT ultra-mass spectrometer (Thermo Fisher Scientific, Schwerte, Germany). Solvent A (water/0.1% formic acid) and solvent B (MeCN/0.1% formic acid) were used at a column temperature of 40 °C and a flow rate of 0.2 mL/min with the following gradient: Linear increase from 2% to 30% B over 18 min, followed by a linear increase from 30% to 95% B in 15 min and keeping 95% B for another 2 min. Absorbance was recorded at 215 nm. In this way, production of all nine MccJ25 was confirmed. For purification, the remainders of the extracts were applied to two rounds of preparative HPLC employing a microbore 1100 HPLC system (Agilent, Waldbronn, Germany) with a VP 250/21 Nucleodur C18 Htec 5 μm column (Macherey-Nagel, Munich, Germany) at room temperature. Solvent A and solvent C (MeOH/0.1% formic acid) were used for the first round, and solvent D (water/0.1% trifluoroacetic acid) and solvent E (MeCN/0.1% trifluoroacetic acid) for the second round of purification. The flow rate was set to 18 mL/min and the absorbance was again detected at 215 nm. A gradient starting with a linear increase from 40% to 55% C in 30 min, followed by a linear increase from 55% to 95% C in 2 min and holding 95% for another 3 min was run for the first purification. The fractions containing the target compounds were identified by MS, dried at 40 °C and reduced pressure, dissolved in 8 mL of 20% MeCN, and then applied to the second round of purification using a gradient starting with a linear increase from 20% to 40% E in 30 min, followed by a linear increase from 40% to 95% E in 2 min and keeping 95% E for another 3 min. Thereby, pure samples of all MccJ25 variants were obtained for further experiments. Yields ranged from moderate (~0.5–0.6 mg/L for 12IGL, 12IGF, and 12VGF) over good (~5–7 mg/L for 13IGL, 13IGF, and 13VGF) to high (~18–25 mg/mL for 11IGL, 11IGF, and 11VGF). The obtained yields are in good agreement with the extent the lasso peptide scaffold was altered: The more the variant sequence differed from the WT sequence, the worse the production became.

### 4.3. Cloning and Protein Purification

Expression constructs for C-terminally Strep-tagged ClpP from *S. aureus* were kindly provided by the group of Prof. S. Sieber (TU Munich) [64]. Expression of C-terminally Strep-tagged ClpP from *S. aureus* was performed in *E. coli* BL21 (DE3). Overnight cultures were transferred into 1 L cultures of LB medium and grown to an optical density at 600 nm (OD_600_) 0.4–0.6 at 37 °C. Expression was induced by the addition of 1 mM isopropyl-β-D-galactopyranosid (IPTG) and cells were then harvested and resuspended in ice-cold lysis/wash buffer (150 mM NaCl, 100 mM Tris-HCl, 1 mM EDTA, pH 8) after 5 h. Cell lysis was performed in a Precellys Homogeniser (Bertin Instruments, Frankfurt am Main, Germany) and the lysate was cleared by centrifugation for 2 h at 20,000× *g* and 4 °C. If necessary, a DNA digest was performed with DNase I. Purification was conducted via affinity chromatography with subsequent size-exclusion chromatography on an ÄKTA Start and ÄKTA Pure system (GE Healthcare, Solingen, Germany), respectively. For affinity chromatography, StrepTrap HP 1 mL columns were used and the protein was eluted with an isocratic gradient of lysis/wash buffer +2.5 mM *d*-Desthiobiotin. The protein fraction was applied to a Superdex 200 HiLoad 16/600 preparation grade size-exclusion column (running buffer: 100 mM NaCl, 20 mM HEPES, pH 7), concentrated in Amicon Ultra Centrifugal Filters (Merck Millipore, Darmstadt, Germany) with a molecular weight cut off of 10 kDa, and stored at −80 °C. Expression constructs and expression strains for enhanced GFP (eGFP) carrying a C-terminal ssrA-tag for ClpXP degradation as well as N-terminally His-tagged ClpX from *S. aureus* with an N-terminal TEV site were kindly provided by the group of Prof. S. Sieber (TU Munich). Expression and purification were performed as described previously [43]. C-terminally His-tagged ClpP from *B. subtilis* was expressed in *E. coli* BL21 (DE3) cells and purified using HisTrap HP 1 mL affinity chromatography columns on an ÄKTA Start system (GE Healthcare, Solingen, Germany). For experiments that required a separation of monomeric and tetradecameric forms of BsClpP, purification was conducted using different buffers with an additional size-exclusion chromatography step where the respective fractions were collected. The exact procedure and the buffers were described previously [43].

### 4.4. Peptide Degradation Assay

Peptide degradation assays were performed in black flat-bottom 96-well plates with a total reaction volume of 100 μL and reaction temperatures of 32 °C and 37 °C for SaClpP and BsClpP, respectively. Stock solutions of ADEP2 and MccJ25 WT or variants were dissolved in DMSO and pre-diluted to 100× of the final concentration. Accordingly, 1 μL of the respective stock solution was placed at the bottom of the wells. For the kinetic assays in which final MccJ25 concentrations of 200 μM were needed, 50 × pre-dilutions were necessary due to a limited solubility of the lasso peptides. 50 μL of a 2× enzyme solution (final assay concentration: 1 μM) in activity buffer was added to the plate and incubated for 15 min at 32 °C and 37 °C for SaClpP and BsClpP activity assays, respectively. The enzyme reaction was started by addition of 49 μL of a solution of the fluorogenic model peptide substrate Suc-Leu-Tyr-AMC in activity buffer at a final concentration of 200 μM. For SaClpP assays, the activity buffer was comprised of 100 mM NaCl and 100 mM HEPES at pH 7. BsClpP activity buffer (50 mM Tris-HCl, 25 mM MgCl_2_, 200 mM KCl, 2 mM DTT, pH 8) was used in BsClpP assays as well as in SaClpP experiments, where the buffer composition was supposed to lead to oligomeric breakdown of the SaClpP tetradecamer to allow testing the activity retaining effect of ClpP agonists. The fluorescence read-out was measured using a Tecan (Crailsheim, Germany) M200Pro plate reader (excitation/emission: 380/460 nm) every 30–60 s for 1 h. Enzyme velocity was determined by linear regression of the initial segment of the fluorescence-time plot in GraphPad Prism 5 (GraphPad Software, San Diego, CA, USA). All assays were performed in triplicate and repeated at least two times.

### 4.5. Casein Degradation Assay

Casein degradation assays were performed analogous to peptide degradation assays. Fluorescein-labelled casein (FITC-casein) was employed at final concentrations of 20 μM in the respective activity buffers (see peptide degradation assay). 1 μL of a 100× stock solution of ADEP2 in DMSO was placed at the bottom of the wells followed by the addition of a 2× SaClpP or BsClpP enzyme solution (1 μM final concentration, unless noted otherwise). The mixture was incubated at 32 °C or 37 °C for 15 min and then the reaction was started by the addition of a 2× FITC-casein solution in the corresponding activity buffer. The fluorescence read-out was measured using a Tecan (Crailsheim, Germany) M200Pro plate reader employing an excitation wavelength of 485 nm and an emission wavelength of 535 nm. Enzyme velocity was acquired by linear regression of the initial segment of the fluorescence graph and plotted against ADEP2 concentration. All assays were performed in duplicate or triplicate.

### 4.6. eGFP Degradation Assay

A reaction mixture of SaClpP (2.8 μM), SaClpX (2.4 μM), eGFP-ssrA (0.36 μM), and an ATP regeneration system (4 mM ATP, 16 mM creatine phosphate, 20 U mL^−1^ creatine phosphokinase) were incubated in PZ buffer (25 mM HEPES, 200 mM KCl, 5 mM MgCl_2_, 1 mM DTT, 10% (*v*/*v*) glycerol, pH 7.6) at 30 °C in a Tecan (Crailsheim, Germany) M200Pro plate reader. The reaction volume was set at 100 μL and the reaction was performed in white flat-bottom 96-well plates. GFP-ssrA was added to the reaction mix after a 10 min pre-incubation and fluorescence was monitored at an emission wavelength of 535 nm after using an excitation wavelength of 465 nm.

### 4.7. Analytical Gel Filtration

Analytical gel filtration chromatography was carried out on an ÄKTA Pure chromatography system (GE Healthcare, Solingen, Germany) with a Superdex 200 3.2 Increase column. Protein samples were diluted 10× in either SaClpP activity (100 mM HEPES, 100 mM NaCl, pH 7) or BsClpP activity buffer (50 mM Tris-HCl, 25 mM MgCl_2_, 200 mM KCl, 2 mM DTT, pH 8). 40 μL of sample were injected into a 10 μL sample loop. Isocratic elution was carried out at a flow rate of 0.075 mL/min and absorption detected at 280 nm. 

### 4.8. 12IGF Stability Assays

To confirm the stability of 12IGF under standard assay conditions (i.e., to exclude proteolysis of 12IGF by ClpP), HPLC-MS analysis was performed on an Agilent 1200 HPLC series equipped with a temperature-controlled sampler and coupled to an Ultra Trap System XCT 6330 (Agilent, Waldbronn, Germany). 5 μL of sample (1 μM SaClpP, 46 μM 12IGF in SaClpP activity buffer) were injected at given time points into a Reprosil-Gold 300 C-18 5 μm column (10 × 2 mm precolumn and 100 × 2 mm separation column, Dr. Maisch HPLC GmbH, Ammerbuch, Germany). Separation was conducted with a linear gradient elution over 55 min from 0% B to 100% B at a flow rate of 400 μL/min. Eluents A and B were composed of 0.1% formic acid in water and 0.06% formic acid in methanol, respectively. Absorption spectra were recorded at wavelengths 220 nm, 260 nm, 280 nm, 360 nm, and 435 nm, while 12IGF showed the strongest signal at 220 nm. Electronspray ionization (ESI) was performed in ultra-scan mode (positive and negative, alternating) with a capillary voltage of 3.5 kV and a drying gas temperature of 350 °C. Data analysis was performed with the software 6300 Series Trap Control Version 6.1 (Bruker Corporation, Billerica, MA, USA).

## Figures and Tables

**Figure 1 ijms-23-00465-f001:**
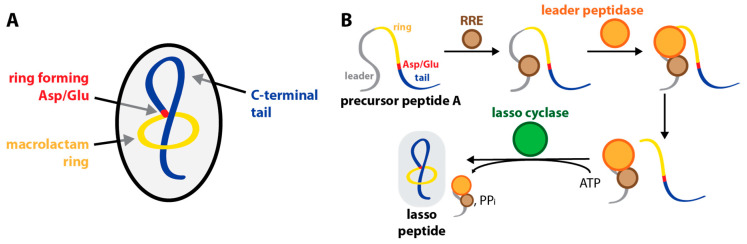
Lasso peptide biosynthesis. (**A**) Schematic showing the key features of the lasso peptide topology. The threaded macrolactam ring is reminiscent of the lariat knot in a lasso, which is the origin of the name of this compound class. Threaded folds can be classified as [x]rotaxanes. In general, the term rotaxane describes mechanically interlocked chemical structures. The number ‘x’ inside the square brackets denotes the number of molecules a rotaxane consists of. Thus, lasso peptides are [1]rotaxanes. (**B**) Schematic depiction of the different steps of lasso peptide biosynthesis as described in the main text. Note that for the biosynthetic machinery producing MccJ25, the RRE protein and leader peptidase are expressed as the single two-domain protein, McjB, that fulfills both of the functions shown here for separate proteins.

**Figure 2 ijms-23-00465-f002:**
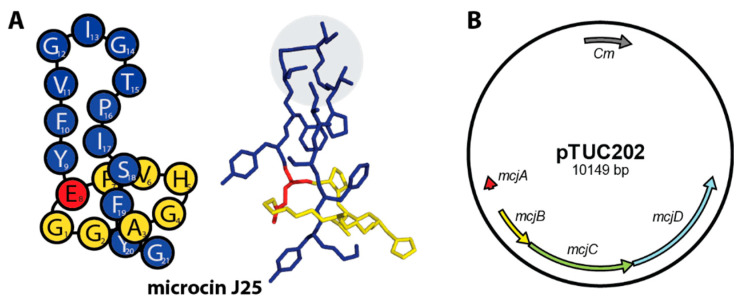
The lasso peptide microcin J25 (MccJ25). (**A**) On the left, a schematic depicting the amino acid sequence of MccJ25. On the right, the NMR structure of MccJ25 (PDB code 1Q71) [26]. The ring forming Glu8 residue is coloured in red, the remaining ring residues in yellow, and the residues of the C-terminal tail in blue. (**B**) The pTUC202 plasmid [21] used for the production of MccJ25 is shown. Variants of MccJ25 can be accessed through mutation of the *mcjA* precursor gene in this plasmid.

**Figure 3 ijms-23-00465-f003:**
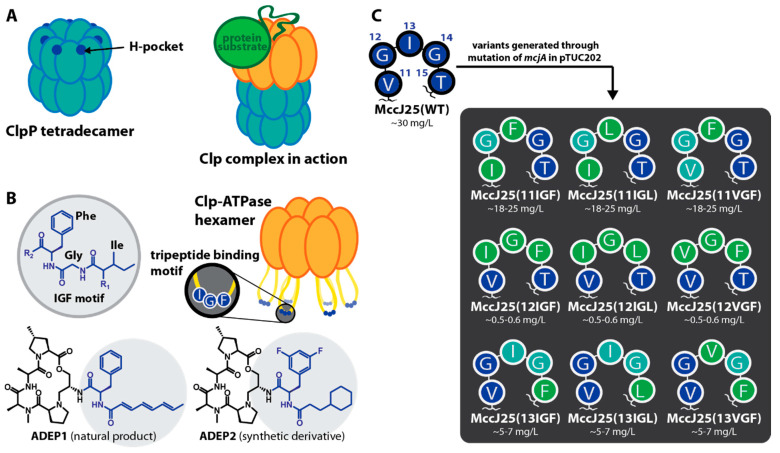
Grafting of the hydrophobic tripeptide binding motifs onto the MccJ25 lasso peptide scaffold. (**A**) Schematic of the ClpP tetradecamer and the complete Clp protease complex, i.e., ClpP in concert with ClpX. The interaction between ClpP and the ClpX ATPase occurs between protruding flexible loops of ClpX, carrying the conserved tripeptide motif (L/I/V)-G-(F/L), and the H-pockets of ClpP. The H-pockets are located between two adjacent monomers of ClpP. In the ClpXP complexes of *S. aureus* and *B. subtilis*, which we used for our biochemical assays, the respective motif is IGF. (**B**) The IGF tripeptide motif of ClpX in comparison with the natural product ADEP1 and the synthetic derivative ADEP2. The latter has increased affinity for SaClpP. The moieties mimicking the IGF motif are highlighted in blue. (**C**) Overview of the generated MccJ25 variants. Three conserved ClpX tripeptide motifs (IGL, IGF, VGF) from different organisms were introduced into the loop region of MccJ25 at three different positions, respectively. The tripeptide motifs in the respective variants are highlighted. Residues of the tripeptide motifs that were introduced by mutagenesis are highlighted in green, while amino acids of the tripeptide motifs that are identical to the corresponding position in WT MccJ25 are depicted in teal. Yields obtained for the variants are indicated.

**Figure 4 ijms-23-00465-f004:**
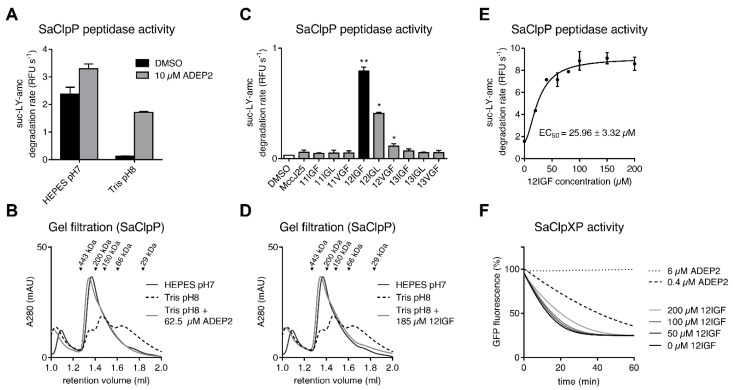
Lasso peptide effects on the Clp system of *S. aureus*. (**A**) Buffer effect on the in vitro peptidase activity of SaClpP. Error bars indicate S.D. (three independent experiments). (**B**) Inactivation by oligomeric breakdown in Tris buffer at pH 8 can be circumvented by the addition of ADEP2. Representative experiment from three repetitions. (**C**) Peptidase activity of SaClpP in Tris buffer at pH 8 in the presence of WT MccJ25 or variants at a concentration of 46 μM. MccJ25 variants carrying the tripeptide motifs at positions 12–14 are the most effective, with 12IGF being the most potent variant. Error bars indicate S.D. (three independent experiments). *, *p* < 0.05; **, *p* < 0.01. (**D**) Dissociation of SaClpP tetradecamers into lower oligomeric states can be prevented by addition of 12IGF. Representative experiment from three repetitions. (**E**) Stabilizing effect of 12IGF on the SaClpP peptidase activity at increasing lasso peptide concentrations with an apparent affinity constant of ~26 μM. Error bars indicate S.D. (three independent experiments). (**F**) Time course of the eGFP-ssrA degradation by SaClpXP in the presence of different concentrations of ADEP2 and 12IGF. 12IGF interferes slightly with SaClpXP activity (three separate experiments, a representative experiment is shown).

**Figure 5 ijms-23-00465-f005:**
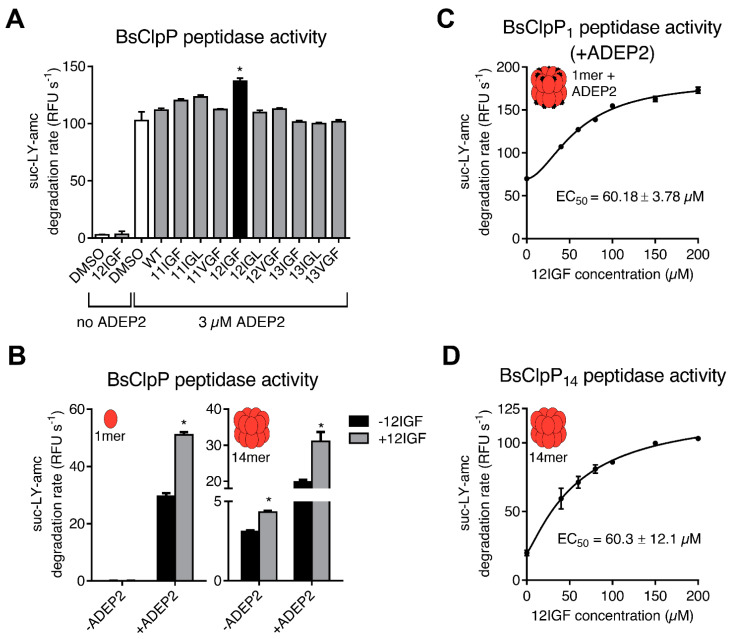
Lasso peptide effect on the Clp system of *B. subtilis*. (**A**) Peptidase activity of BsClpP in Tris buffer at pH 8 in the presence or the absence of 3 μM ADEP2 and the additional presence of WT MccJ25 or variants thereof at a concentration of 23 μM. The used protein preparation was generated by a procedure widely used for BsClpP [43] and contained predominantly monomers. Error bars indicate S.D. (three independent experiments). *, *p* < 0.05. (**B**) Peptidase activity of separate pre-purified fractions of either the BsClpP monomer (left) or the BsClpP tetradecamer (right) with different combinations of effectors (effector concentrations: 3 μM (ADEP2), 46 μM (12IGF)). Error bars indicate S.D. *, *p* < 0.05. (**C**) Peptidase activity of the monomeric BsClpP in the presence of 5 μM of ADEP2 and 12IGF at increasing concentrations. Error bars indicate S.D. (two separate experiments). (**D**) Peptidase activity of the pre-purified tetradecameric BsClpP at increasing 12IGF concentrations and in the absence ADEP. Error bars indicate S.D. (two separate experiments).

**Figure 6 ijms-23-00465-f006:**
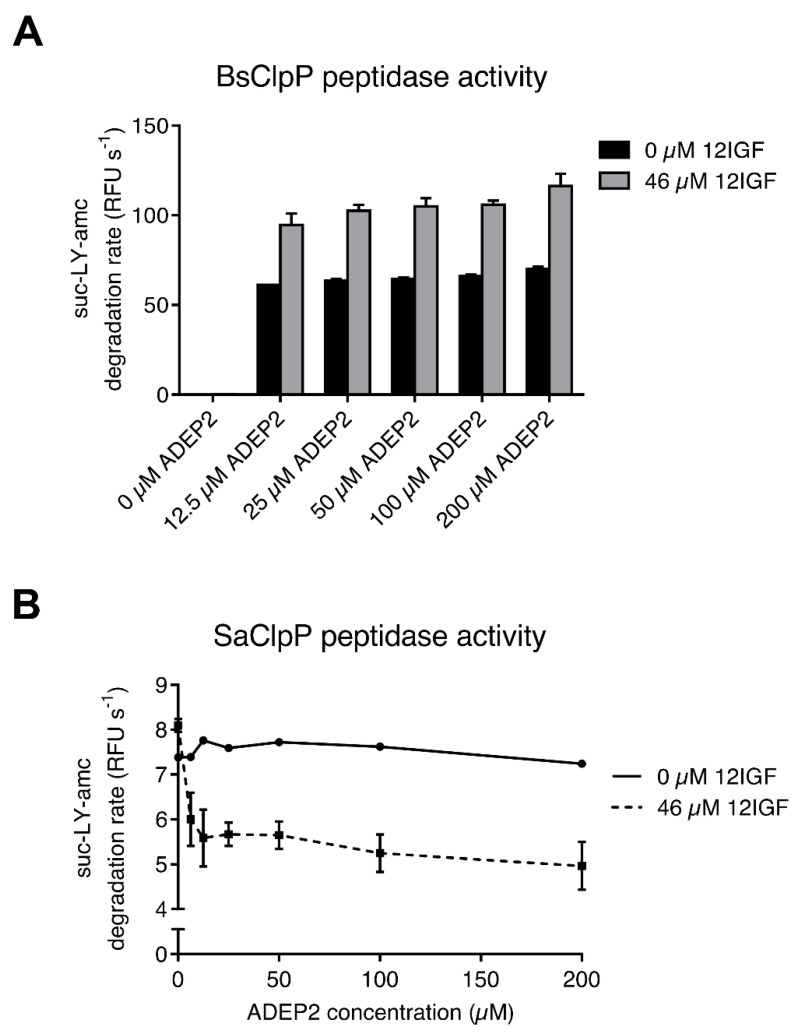
ADEPs and 12IGF show noncompetitivity. (**A**) BsClpP in vitro peptide degradation assays in the absence and the presence of 12IGF at increasing ADEP2 concentrations. The synergistic effect of 12IGF remains stable even at high ADEP2 concentrations. Error bars indicate S.D. (three separate experiments). (**B**) SaClpP in vitro peptide degradation in the absence and the presence of 12IGF at increasing ADEP2 concentrations. The combination of ADEP2 and 12IGF leads to a decrease of peptide degradation activity of SaClpP. No converging trend at increasing ADEP2 concentrations can be observed. Error bars indicate S.D. (two separate experiments).

**Table 1 ijms-23-00465-t001:** SLIM primers for mutagenesis of *mcjA* in the pTUC202 plasmid. SLIM overhangs are underlined and mutated bases are highlighted in bold.

Name	Sequence
*mcjA_SLIM_FP*	TTC TAT GGC TGA TAT TCT GAA AGA AGA ACT CTG
*mcjA_SLIM_RP*	CTC AGG CAC ATG TCC TGC ACC ACC
*mcjA_SLIM_11IGL_Tail_FP*	TAT TTT **ATT** GGG **CTG** GGT ACA CCT ATA TCT TTC TAT GGC TGA TAT TCT GAA AGA AGA ACT CTG
*mcjA_SLIM_11IGL_Tail_RP*	AGA TAT AGG TGT ACC **CAG** CCC **AAT** AAA ATA CTC AGG CAC ATG TCC TGC ACC ACC
*mcjA_SLIM_11IGF_Tail_FP*	TAT TTT **ATT** GGG **TTT** GGT ACA CCT ATA TCT TTC TAT GGC TGA TAT TCT GAA AGA AGA ACT CTG
*mcjA_SLIM_11IGF_Tail_RP*	AGA TAT AGG TGT ACC **AAA** CCC **AAT** AAA ATA CTC AGG CAC ATG TCC TGC ACC ACC
*mcjA_SLIM_11VGF_Tail_FP*	TAT TTT GTG GGG **TTT** GGT ACA CCT ATA TCT TTC TAT GGC TGA TAT TCT GAA AGA AGA ACT CTG
*mcjA_SLIM_11VGF_Tail_RP*	AGA TAT AGG TGT ACC **AAA** CCC CAC AAA ATA CTC AGG CAC ATG TCC TGC ACC ACC
*mcjA_SLIM_12IGL_Tail_FP*	TAT TTT GTG **ATT GGC CTG** ACA CCT ATA TCT TTC TAT GGC TGA TAT TCT GAA AGA AGA ACT CTG
*mcjA_SLIM_12IGL_Tail_RP*	AGA TAT AGG TGT **CAG GCC AAT** CAC AAA ATA CTC AGG CAC ATG TCC TGC ACC ACC
*mcjA_SLIM_12IGF_Tail_FP*	TAT TTT GTG **ATT GGC TTT** ACA CCT ATA TCT TTC TAT GGC TGA TAT TCT GAA AGA AGA ACT CTG
*mcjA_SLIM_12IGF_Tail_RP*	AGA TAT AGG TGT **AAA GCC AAT** CAC AAA ATA CTC AGG CAC ATG TCC TGC ACC ACC
*mcjA_SLIM_12VGF_Tail_FP*	TAT TTT GTG **GTG GGC TTT** ACA CCT ATA TCT TTC TAT GGC TGA TAT TCT GAA AGA AGA ACT CTG
*mcjA_SLIM_12VGF_Tail_RP*	AGA TAT AGG TGT **AAA GCC CAC** CAC AAA ATA CTC AGG CAC ATG TCC TGC ACC ACC
*mcjA_SLIM_13IGL_Tail_FP*	TAT TTT GTG GGG ATT GGT **CTG** CCT ATA TCT TTC TAT GGC TGA TAT TCT GAA AGA AGA ACT CTG
*mcjA_SLIM_13IGL_Tail_RP*	AGA TAT AGG **CAG** ACC AAT CCC CAC AAA ATA CTC AGG CAC ATG TCC TGC ACC ACC
*mcjA_SLIM_13IGF_Tail_FP*	TAT TTT GTG GGG ATT GGT **TTT** CCT ATA TCT TTC TAT GGC TGA TAT TCT GAA AGA AGA ACT CTG
*mcjA_SLIM_13IGF_Tail_RP*	AGA TAT AGG **AAA** ACC AAT CCC CAC AAA ATA CTC AGG CAC ATG TCC TGC ACC ACC
*mcjA_SLIM_13VGF_Tail_FP*	TAT TTT GTG GGG **GTG** GGT **TTT** CCT ATA TCT TTC TAT GGC TGA TAT TCT GAA AGA AGA ACT CTG
*mcjA_SLIM_13VGF_Tail_RP*	AGA TAT AGG **AAA** ACC **CAC** CCC CAC AAA ATA CTC AGG CAC ATG TCC TGC ACC ACC

**Table 2 ijms-23-00465-t002:** Composition of 500X M9 vitamin mix.

Component	Amount
choline chloride	1.0 g
folic acid	1.0 g
pantothenic acid	1.0 g
Nicotinamide	1.0 g
myo-inositol	2.0 g
pyridoxal hydrochloride	1.0 g
Thiamine	1.0 g
Riboflavin	0.1 g
disodium adenosine 5′-triphosphate	0.3 g
Biotin	0.2 g
	add 300 mL ddH_2_O *

* After addition of ddH_2_O, a solution of 10 M NaOH was slowly added until a clear solution was obtained (the final pH is usually around 12). After sterile filtration of the clear vitamin mix, it was either stored for short-term at 4 °C or for long-term at −20 °C.

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
