# Peer review of "Generation of Lasso Peptide-Based ClpP Binders"

_ijms, 2021, doi:10.3390/ijms23010465_

Round 1
Reviewer 1 Report
The manuscript of Malik et al subjected to my revision describes the utilization of lasso peptide mutants in order to design new agents interacting with Clp protease. This is an interesting, well-written paper. It has an extensive methodology section, the intention of the authors is well-justified, and the results are sufficiently described and in my opinion, the right conclusions are drawn. I particularly like the approach of the authors, who honestly admit, that they expected different results ("This came as a surprise since the MccJ25 variants had been designed specifically to 333 mimic the Clp-ATPase IGF-loop binding.").
Following this, I have only minor remarks:
- The caption for Fig. 1. I do not agree with the sentence "The threaded macrolactam ring is reminiscent of the knot in a lasso". The knots in proteins are very different things than lassos. According to my knowledge, lasso peptides have nothing to do with cases, where the protein backbone is tied into a (mathematical) knot.
- Page 3, line 90 - as far as I understand, after detaching the tail from the loop, the resulting structure has still the topology of [1]rotaxane, not [2] rotaxane, as it is written.
- Fig. 4 - panels B and D show two identical traces. Maybe they could be joined together to show, how the influence of the two factors differs?
- Page 11, line 363 - isn't the affinity constant of 12IGF LOWER for than for ADEPs? In the particular line, it is "higher".
- Page 11, line 373 - the authors claim, that the effect is due to the tripeptide motif, not secondary effects, but they also show, that the same tripeptide motif placed differently (e.g. 11IGF) has a significant effect on the mutant activity.
Also, the thing which I miss in the paper is some comparison of the stability of different factors. One of the ideas to use lasso peptides is to utilize their exceptional stability, so it is somehow expected to study the stability-related advantages of such factors. But I expect the authors will perform such experiments in the follow-up works.
Taking all the things into account, I recommend publication of the manuscript after responding to the minor remarks stated above.
Reviewer 2 Report
In this study, the authors engineered microcin J25 by introducing the conserved tripeptide motifs into the lasso peptide loop region to mimic the Clp-ATPase loops. They found 12IGF conferred stability to ClpP tetradecamers and stimulated peptide degradation, but it did not trigger unregulated protein degradation. They also observed synergistic interactions between 12IGF and ADEP. This study is well designed and presented. It’s a very nice work. I have no improving comments on it.
